# Moderate beta-cell ablation triggers synergic compensatory mechanisms even in the absence of overt metabolic disruption

Andreas Frøslev Mathisen[1], Ulrik Larsen[1], Natalie Kavli [1], Lucas Unger[1], Laura Maria Daian[2], Andrei Mircea Vacaru[2], Ana-Maria Vacaru[2], Pedro Luis Herrera[3], Luiza Ghila [1] & Simona Chera [1] ✉

Regeneration, the ability to replace injured tissues and organs, is a phenomenon commonly associated with lower vertebrates but is also observed in mammals, in specific tissues. In this study, we investigated the regenerative potential of pancreatic islets following moderate beta-cell loss in mice. Using a rapid model of moderate ablation, we observed a compensatory response characterized by transient inflammation and proliferation signatures, ultimately leading to the recovery of beta-cell identity and function. Interestingly, this proliferative response occurred independently of inflammation, as demonstrated in ablated immunodeficient mice. Furthermore, exposure to high-fat diet stimulated beta-cell proliferation but negatively impacted beta-cell function. In contrast, an equivalent slower ablation model revealed a delayed but similar proliferative response, suggesting proliferation as a common regenerative response. However, high-fat diet failed to promote proliferation in this model, indicating a differential response to metabolic stressors. Overall, our findings shed light on the complex interplay between beta-cell loss, inflammation, and stress in modulating pancreatic islet regeneration. Understanding these mechanisms could pave the way for novel therapeutic strategies based on beta-cell proliferation.

Regeneration is the capacity of replacing injured structures, such as tissue and organs, after birth. Despite common beliefs, mammals can deploy a wide range of regenerative responses in certain contexts[1]. The extent of these regenerative responses are however limited, lacking the patterned 3D reconstruction observed in classical regeneration systems, and thus falling under the umbrella of tissue repair regenerative programs[2].

Following stress or injury, mice can regenerate tips of digits[3,4], liver[5], and pancreatic tissue[6–8], as well ass the cochlea[9] (neonatally), to name some. The murine pancreas has an unexpected plasticity potential, being able to deploy a wide range of responses following the insulin-secreting beta-cell destruction[10]. Similar to other regenerating mammalian tissues, the regenerative program deployed by the pancreatic islet is dependent on the type of insult, the age at time of injury, and extent of beta-cell loss[11]. Briefly, there are two main regeneration mechanisms employed by the murine beta-cells: (1) transdifferentiation (direct[8,12] or indirect[7]) following total beta-cell loss and

(2) proliferation of the surviving beta-cells[13–17], following subtotal, yet extensive beta-cell damage[18–23]. Both mechanisms occur naturally in response to diverse stressors but are limited and inefficient, especially in adults. However, understanding the cellular and molecular basis of these regenerative programs is important for their subsequent fostering and boosting.

Consequently, a number of model systems of beta-cell loss has been developed, ranging from chemical (streptozotocin (STZ)[24–28], alloxan[29,30]) to genetic (DTR[8], DTA[13,31]) ablation. These models have distinct characteristics and efficiencies of ablation and can be employed to study the islet regeneration potential in response to different extents of beta-cell loss. However, as in homeostatic conditions 10-14% beta-cell mass is sufficient to maintain normoglycemia[32], most studies aim for an extreme beta-cell destruction to simulate insulin deficient diabetes, thus putting the islet under metabolic pressure to regenerate new insulin cells.

[1]Mohn Research Center for Diabetes Precision Medicine, Department of Clinical Science, University of Bergen, Bergen, Norway. [2]BetaUpreg Research Group, Institute of Cellular Biology and Pathology "Nicolae Simionescu", Bucharest, Romania. [3]Department of Genetic Medicine and Development, Faculty of Medicine, University of Geneva, Geneva, Switzerland. ✉e-mail: Simona.Chera@uib.no

In these contexts, it is rather unclear if the earliest signal initiating the regeneration program is extra[33]/intra-islet[6] insulin demand, or if it is an innate response triggered by dying beta-cells or subsequent tissue remodeling[34]. Identifying the type of regeneration trigger is essential for further improving regeneration potential and efficiency.

Here we addressed this knowledge gap by employing models of beta-cell loss allowing the specific ablation of only 50% of beta-cells and investigated the regenerative response in the absence of a metabolic demand for insulin. In comparison with the previous models of total (99.98%) and subtotal (80–90%) beta-cell ablation, the loss of ~50% beta-cells is moderate, with the surviving beta-cell mass being well above the required physiological threshold. Moreover, as the beta-cell regeneration mechanism depends on the residual beta-cell mass, it will be important to establish the type of response to beta-cell loss, if any. At least two regenerative mechanisms were described in models of massive ablation, with total destruction (99.98%) leading to the transdifferentiation from neighboring endocrine cell types[6–8,12], while subtotal loss (80-90%) triggering a proliferative response of the surviving beta-cells[13–15,35,36]. We thus further explored the relationship between the observed regeneration mechanisms, inflammation and additional metabolic stressors, according to the type of beta-cell ablation system and their specific ablation dynamic.

## Results

### Genetic ablation of hemizygous RIP-DTR females leads to an average 50% insulin-secreting beta-cells loss without impacting glucose regulation

In order to establish if subtotal ablation of insulin-secreting beta-cells leads to a regenerative response, we employed the RIP-DTR[8] (rat insulin

promoter—diphtheria toxin receptor) system. The DTR system is based on the fact that wild-type mice have a negligible expression of DTR (diphtheria toxin receptor). Thus, ectopic DTR expression in a cell type of interest, such as insulin-secreting beta-cells, will specifically render the cell sensitive to toxin action and subsequent death by apoptosis. In the case of the RIP-DTR model, the DTR is under the control of RIP (rat insulin promoter) ensuring its specific expression in insulin-expressing cells. Of importance, the transgene is knocked in the *Hprt* locus on the X-chromosome. Due to the inherent random X-inactivation phenomena, in hemizygous female ($X^{hprt}X^{DTR}$) only between 40–60% of the insulin-expressing beta cells will express the DTR, being targeted by DT and subsequent apoptosis. Consequently, this system allows achieving very reproducible, rapid and specific overall subtotal (50%) ablation (hemi-ablation) efficiency in only few days post ablation (DPA) (Fig. 1a).

As expected, ablated hemizygous females remained normoglycemic (Fig. 1b), proving that indeed 50% beta-cell mass was sufficient for controlling glycemia at least during normal metabolic load. Moreover, both moderately ablated (50%) and non-ablated females displayed largely similar glucose tolerance as revealed by IPGTT (intra peritoneal glucose tolerance test), further confirming the capacity of hemi-ablated females to cope with the metabolic demand (Fig. 1c). We further analysed if the pancreatic islet would deploy a regenerative response, without an obvious metabolic pressure for insulin, i.e., in the absence of a clear regenerative demand.

### Rapid moderate ablation of insulin-secreting beta-cells leads to transcriptional changes compatible with regeneration

To investigate the cellular and molecular changes occurring following subtotal (50%) ablation in these mice, we designed a regenerative timeline of

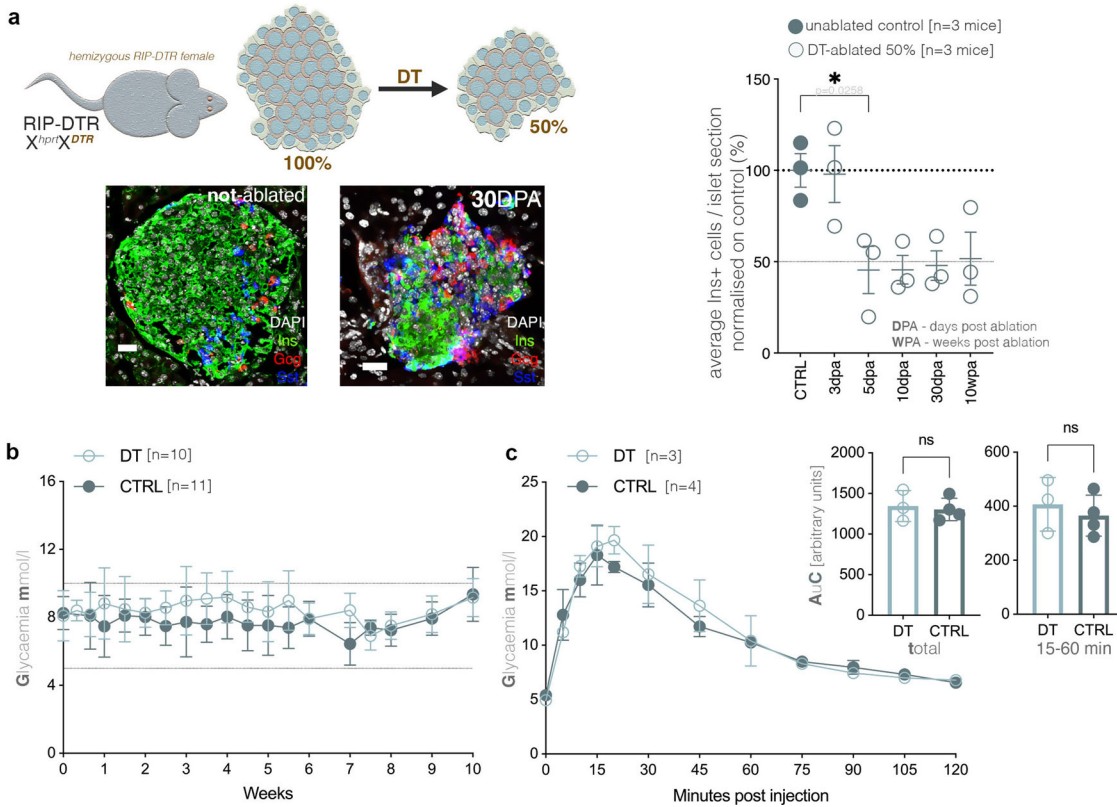

**Fig. 1 | Characterization of the hemizygous RIP-DTR females following moderate DT-induced ablation of beta-cells. a** Schematic illustration and graph depicting the timeline for quantification of moderate ablation using hemizygous RIP-DTR females, including representative immunofluorescence images of DAPI (white), insulin (green), glucagon (red), and somatostatin (blue), before and after ablation (scale bar - 20 μm) and graph depicting the ablation efficiency (one-way ANOVA, each data point represents one distinct mouse, an average of 55 islet sections were assessed / mouse, DPA – days post ablation; WPA- weeks post ablation). **b** Glycaemic curves of DT-treated (*n* = 10) and unablated controls (*n* = 11), over 10 weeks post ablation (WPA). **c** Graphs depicting results of IPGTT at 10 WPA in DT-ablated (*n* = 3) and control (*n* = 4) mice (two-way ANOVA with Bonferroni correction) as well as comparisons of AuCs (area under curve) for the total IPGTT course and the 15 to 60 min subperiod. Data are represented as mean ± SEM (standard error of the mean).

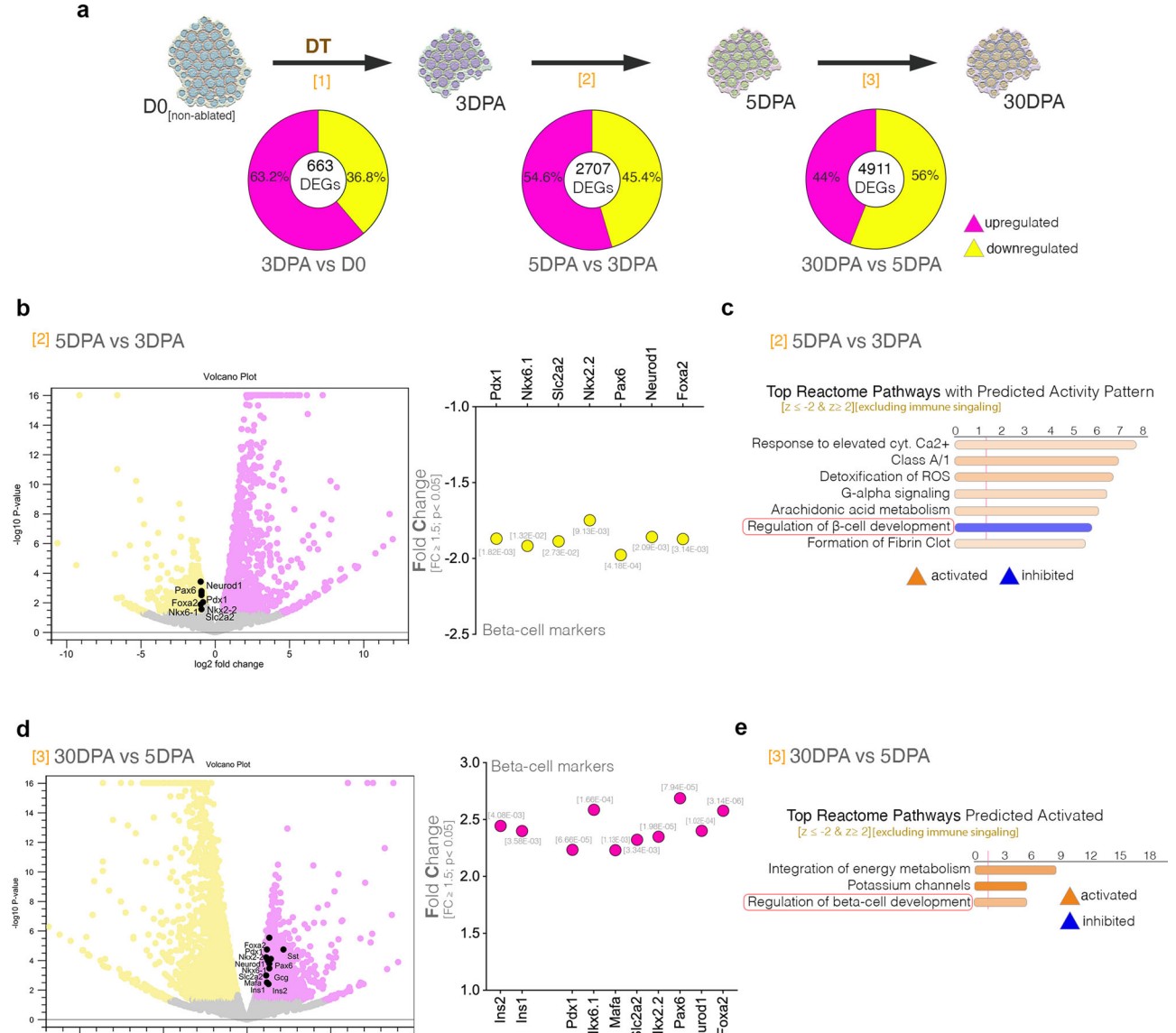

**Fig. 2 | Dynamic characterization of the islet transcriptional signature in the first month post-moderate ablation. a** Graphic illustration of the four compared timepoints along with pie diagrams describing the number of differentially expressed genes (DEGs, FC ≥ 1.5, $p$ < 0.05) and listed percentages of upregulated and downregulated DEGs (one-way ANOVA). **b** Volcano plot and graph displaying the observed downregulation of beta-cell markers between 5 and 3 DPA, $p$-values are listed below bars. **c** Top predicted Reactome pathways with predicted activity pattern between 5 and 3 DPA. (z-score ≤ −2 [inhibited, blue], z-score ≥ 2 [activated, orange]), excluding the immune signaling pathways. **d** Volcano plot and graph displaying the observed upregulation of beta-cell markers between 30 and 5 DPA, $p$-values are listed above bar. **e** Top predicted differentially regulated Reactome pathways between 30 and 5 DPA. (z-score ≤ −2 [inhibited, blue], z-score ≥ 2 [activated, orange]), excluding immune signaling pathways, $p$-values are listed above bar.

the early regeneration time-points (from 3 to 30 DPA) and characterized by next-generation sequencing and immunofluorescence microscopy. The transcriptome comparison of the different regenerative intervals revealed that the number of differentially expressed genes (DEGs, FC ≥ 1.5, $p$ < 0.05) is increasing with time (from 663 DEGs characterizing the 1st interval to 4911 DEGs in the 3rd), suggesting that the 50% ablation is not a transcriptionally passive process (Fig. 2a). Moreover, the upregulated and downregulated DEGs were represented in largely similar fractions, suggesting that the changes in the transcriptional profile are not just caused by the loss of cells.

To identify the changes in the transcriptional profile characterizing each regenerative interval, we performed pathway analysis. During the first three days following DT administration (0 to 3 DPA – 1st interval), the analysis revealed the activity of pathways involved in extracellular matrix organization and collagen fiber remodeling (Supplementary Fig. 1a).

Consistently, cellular movement, molecular transport and cell-to-cell signaling were the top 3 molecular and cellular functions, while endocrine system disorder was the top disorder pinpointed by the analysis (Supplementary Fig. 1b). Overall, these data are consistent with a tissue remodeling phase characterizing the first three days following moderate beta-cell ablation.

For the next interval (3 to 5 DPA following DT administration, 2nd interval) the analysis revealed an increased activity of the pathways involved in inflammation (Supplementary Fig. 1c), with inflammatory response being inferred the top disorder compatible with the analyzed transcriptional landscape (Supplementary Fig. 1d). Moreover, the immunofluorescence (IF) and quantification of key immune cells markers such as CD45 (marker for all nucleated hematopoietic cells) also confirmed a significant increase of Cd45+ cells within the islet at 5 DPA (Supplementary Fig. 1e).

Importantly, key markers of beta-cell identity (such as *Pdx1, Nkx6.1* amongst others) and functionality (such as *Slc2a2*, Glut2, the main glucose transporter in mice) were significantly downregulated during this period (Fig. 2b) consistent with beta-cell loss. This was further substantiated by the inferred inhibition of Regulation of beta-cell development signaling in the top reactome pathways (Fig. 2c, red frame). These results suggest that between 3 and 5 DPA the apoptotic beta-cells are removed, leading to a steep decrease in beta-cell markers.

Finally, the longer and last interval analyzed (5 to 30 DPA, 3rd interval) was characterized by a decrease of the previously observed inflammatory signature (Supplementary Fig. 1f). Interestingly, during this period both murine insulin genes (*Ins1* and *Ins2*) and a wide range of key beta-cell markers (*Pdx1, Nkx6.1, Mafa, Slc2a2*) were observed significantly upregulated (Fig. 2d), suggesting a compensatory response. Consistently, the Regulation of beta-cell development was now inferred as activated by the pathway analysis (Fig. 2e, red frame). Similarly, the integration of energy metabolism was predicted as activated, indicating an improvement of the energy balance at 30 DPA, as compared to the earlier point (Fig. 2e). Surprisingly, two other key islet hormones, somatostatin (*Sst*) and glucagon (*Gcg*) were observed upregulated (Supplementary Fig. 1g). These data indicate a recovery of the beta-cell markers levels during the first month post-ablation, however, it is unclear whether this is caused by a potential regenerative event, a functional rebound or both.

Globally the analysis of the early regenerative line following moderate DT ablation revealed an initial stage of tissue remodeling, followed by a brief inflammatory signature, peaking at 5 DPA, doubled by a decrease of key beta-cell markers. The beta-cell markers decrease was transient, their expression levels returning by 30 DPA, reaching levels equal to or even higher (as for *Ins1*) than unablated control (Supplementary Fig. 1h), indicating the occurrence of compensatory mechanisms.

## Beta-cell proliferation is transiently increased following rapid moderate ablation

We further investigated the type of compensatory response deployed following hemi-ablation and initially assessed the occurence of the alpha-to-beta cell conversion, the main regenerative mechanism following total (99%) DT-induced beta-cell ablation. IF indicated no significant increase in bihormonal cells, suggesting that alpha-cell conversion does not contribute to the observed compensatory response (Supplementary Fig. 1i)

We, then, analyzed the beta-cell proliferative response along the regenerative timeline. Interestingly, the pathway analysis identified important signaling involved in cell cycle progression in the top pathways of the second period (i.e., 3 DPA to 5 DPA), all inferred as activated (Fig. 3a). Indeed, key proliferation markers such as *Mki67* (Ki67), *Foxm1, Aurka* and *Pcna* were observed upregulated at 5 DPA as compared to 3 DPA (Fig. 3b). Surprisingly, the senescence marker *Cdkn1a* (p21) was also observed upregulated (Fig. 3b, volcano plot). Of note, the key proliferation markers analyzed were also observed significantly upregulated when compared to the control, non-ablated, islets (Supplementary Fig. 1j), indicating also an increase in proliferation as compared to homeostatic levels.

Notably, the activity pattern observed between 3 DPA and 5 DPA after ablation was transient, being inferred inhibited during the third period (i.e., 5 DPA to 30 DPA, Fig. 3c). As expected, during this last period the same proliferation markers (*Mki67, Foxm1, Aurka* and *Pcna*) were observed downregulated at 30 DPA as compared to 5 DPA (Fig. 3d). Moreover, the cyclin-dependent kinase inhibitors, *Cdkn1a* (p21), *Cdkn2c* (p18-INK4c) and *Cdkn3* (Cip2), followed the same regulatory trend (Fig. 3d, volcano plot).

The observed transient increase in beta-cell proliferation was confirmed also by Ki67 protein staining and quantification, with the percentage of Ins + Ki67+ cells per islet section more than doubled (2.78x increase) between 5 DPA and 3 DPA (Fig. 3e, $p = 0.0175$). In contrast, the alpha- and delta-cells did not display a significant proliferative response (Supplementary Fig. 1k).

Overall, these data indicate a transient proliferation event peaking at 5 DPA, consistent with a regeneration-based compensatory event. Confirmatory, the comparison of large size-similar islets (an average of 140 cells/islet section) over time indicated an increase of Insulin+ cells ratio at 10 WPA as compared to 5 DPA (Fig. 3f, $p = 0.0035$).

## The inflammatory signature is not a prerequisite for beta-cell proliferation

To investigate the causal relationship between the increase in the inflammatory signature and proliferation index at 5 DPA, we performed the moderate ablation (50%) in a distinct NSG (NOD scid gamma) immunodeficient RIP-DTR model (Fig. 3g). These mice are extremely immunodeficient due to two mutations: *scid* (*Prkdc* mutation) and *IL2rg^null* on a NOD genetic background. They are deficient in mature T and B lymphocytes (*scid*) and NK cells' cytotoxic activity is very low (*IL2rg^nul*). Of note, in contrast with NOD mice, NSG mice do not develop spontaneous diabetes.

The comparison pathway analysis of the early regeneration timeline in the NSG RIP-DTR indicated the activation of the cell cycle-related signaling in 5 DPA islets when compared to 3 DPA islets (Fig. 3h, pink frame), similar to its immunocompetent counterpart. Consistently, key proliferation markers like *Mki67, Foxm1*, and *Aurkb* were upregulated at 5 DPA as compared to 3 DPA, also in the immunodeficient mice (Fig. 3i, Supplementary Fig. 2a), suggesting that lymphocytes and NK cells do not play a crucial role in the process.

As the NSG mice still retain neutrophiles and macrophages, we further investigated if these may be involved in a potentially inflammatory signature promoting the observed proliferative response in the ablated NSG mice by datamining the relevant datasets (5DPA vs 0DPA and 5DPA vs. 3DPA) for specific markers. The analysis revealed the decrease or no significant regulation of key murine neutrophile markers, such as *Ly-6g, Cd18 (Itgb2)* or *Cd49 (Itga1-6), Cd15 (Fut4)* and *Cd16(Fcgr3)* (Supplementary Fig. 2b).

In contrast, we identified a significant increase in major pan-macrophage markers such as *Cd68*, and *Cx3cr1* and *F4/80 (Adgre1)* (Fig. 3j). We further explored if the involved macrophages belong to the M1 class (pro-inflammatory) or M2 class (anti-inflammatory). The analysis of key markers differentiating M1 and M2 macrohpages[37] revealed that M1-specific markers, like *Cd38*[37] and also iNOS (*Nos2*), *CD86, CD80, Tnf, Gpr18, Fpr2* were unchanged (Supplementary Fig. 2c). In contrast M2-specific markers, such as Egr2, displayed a significant upregulation, with a peak at 3DPA (Fig. 3k). These results indicate the presence of the M2 anti-inflammatory macrophages in the islet at the time proliferation is increased. Overall, these data suggest that inflammation is not necessary for mounting the observed increase in proliferation, yet the immune system is probably involved in tuning the process.

## Diet stress promotes beta-cell proliferation, without improving their identity and function

We further attempted to promote the regeneration potential by increasing the functional pressure on the islet using high fat diet (HFD), as this approach was previously demonstrated to induce beta-cell proliferation[38]. In this experiment, the mice recovered for ten weeks either under HFD or synthetic diet (SD – the recommended control diet for HFD experiment), with the DT administration and dietary switch occurring at the same time (Fig. 4a). The mice exposed to HFD did not exhibit a significant change in glycemia up until ten weeks as compared to the control SD counterparts, regardless of the ablation status (Fig. 4b, Fig. 4c). Moreover, at ten weeks, the IPGTT did not reveal significant changes in glucose stimulated insulin secretion neither in control nor in DT mice exposed to HFD (Fig. 4d).

The pathway analysis inferred the activation of a large number of cell cycle signaling in the ablated islets exposed to HFD, when compared with the ones exposed to SD (Fig. 4e). As expected, key cell cycle markers were strongly upregulated in HFD-ablated islets (Fig. 4f), suggesting that the

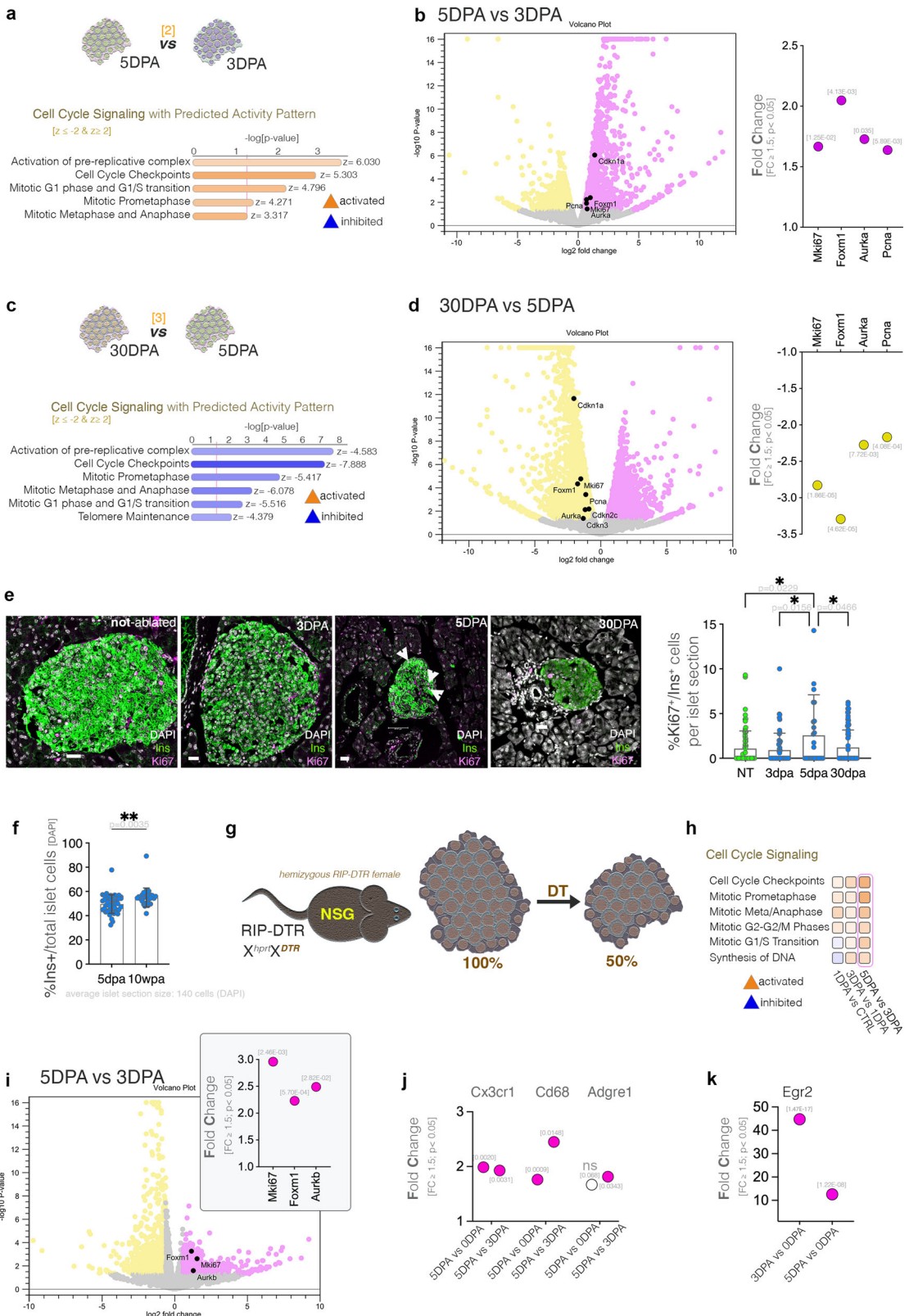

metabolic stressor promotes proliferation in moderately ablated islets. Interestingly, despite this proliferation increase, the insulin genes as well as key beta-cell identity and functionality markers are observed significantly downregulated in the ablated islets exposed to HFD (Fig. 4g).

The data suggest that, while further metabolic stress promotes islet proliferation, it however fails to mount a proper regenerative response.

## Slowly progressing, moderate beta-cell ablation exhibits a different response dynamic

To investigate if the observed phenomena characterize a general response of the islet to the subtotal (50%) ablation, we employed a model of chemical beta-cell ablation. In these mice, beta-cells are targeted and slowly destroyed using streptozotocin (STZ), a chemical agent able to reach beta-cells using

**Fig. 3 | Assessment of the proliferation activity in the surviving beta-cells.**
**a** Predicted activity of the cell cycle signaling pathways between 5 and 3DPA (z-score ≤ -2 [inhibited, blue], z-score ≥ 2 [activated, orange]). **b** Volcano plot and graph displaying the observed upregulation of proliferation markers in 5 DPA as compared to 3 DPA, p-values are listed above bar. **c** Predicted regulation of cell cycle signaling pathways between 30 and 5 DPA (z-score ≤ -2 [inhibited, blue], z-score ≥ 2 [activated, orange]). **d** Volcano plot and graph displaying the observed down-regulation of proliferation markers at 30 DPA compared to 5 DPA, p-values are listed above bar. **e** Representative immunofluorescence images of DAPI (white), insulin (green) and Ki67 (magenta) (scale bar – 20 μm) and graph displaying the quantification of Ki67+Ins+ cells per islet section (mean ± SEM, one-way ANOVA; N = 3 mice, an average of 68 islet sections were counted per condition, arrowheads at 5DPA point at Ki67+ cells). **f** Procentage of Ins+ cells from total islet cells (DAPI)

(unpaired t-test with Welch's correction; an average of 30 large size-similar islets (an average of 140 cells/islet section) were counted per condition). Data are represented as mean ± SEM. **g** Schematic illustration of the moderate ablation set-up in the immunodeficient RIP-DTR hemizygous NSG female mice. **h** Comparison analysis depicting the activity pattern of cell cycle pathways in moderately ablated NSG mice over the first five days post ablation (1, 3 and 5 DPA) (z-score ≤ -2 [inhibited, blue], z-score ≥ 2 [activated, orange]). **i** Volcano plot and graph displaying the observed regulation of proliferation markers when comparing islets from NSG mice at 5 DPA and 3 DPA, p-values are listed above bar. **j** Graph displaying the observed regulation of macrophage markers between 5 DPA and 0 DPA as well as 5 DPA and 3 DPA in NSG mice, p-values are listed above bar. **k** Graph displaying the observed regulation of the M2 macrophage marker Egr2 between 3 DPA and 0 DPA as well as 5 DPA and 0 DPA in NSG mice, p-values are listed above bar.

the Glut2 (*Slc2a2*) glucose transporter[39]. Of note, STZ is also associated with kidney toxicity[40,41], while the ablation efficiency is less reproducible than in the RIP-DTR model.

We initially explored the STZ dose concentration necessary to achieve an equivalent ablation efficiency to the one obtained in hemizygous females, i.e. 50%. By performing a dose response experiment, we identified that a single 75 mg/kg, or 100 mg/kg dose leads to a partial ablation of ~50% at 4-weeks post injection (Fig. 5a). Like the moderately-ablated RIP-DTR counterparts, the STZ-treated mice were all normoglycemic over 30 DPA, however they exhibited significantly impaired glucose stimulated insulin secretion even ten weeks after the injection (Fig. 5b). Moreover, in contrast to DT-ablated animals the maximum ablation efficiency was achieved significantly slower, with the lowest number of Ins+ cells per islet section being detected at 30 days post STZ administration (Fig. 5c). Interestingly, we detected a significant increase in the proportion of beta-cells positive for the proliferation marker Ki67, at 30 days post STZ administration (Fig. 5d). This indicates that the increase in proliferation is retained in STZ-ablated islets, yet it occurs significantly later than following DT ablation (5 days in DT-ablated vs 30 days in STZ-ablated, compare Fig. 3d and Fig. 5d).

Overall, these data indicate that, as expected, the slow STZ-ablation follows a different dynamic, nevertheless a transient increase in proliferation is also observed in this model.

## In contrast with the DT model, the diet stress does not promote beta-cell proliferation in the STZ-induced moderate ablation

To investigate if, as in the DT-model, the HFD, as an additional stressor, can boost cell proliferation in the STZ-induced moderate ablation, we exposed the mice to HFD and SD starting with the first day of STZ treatment for ten weeks and used pathway analysis as readout for the changes in the transcriptional landscape. This analysis indicated metabolic changes as the main difference in the transcriptional landscape between the HFD and SD STZ-ablated islet. A large number of metabolic signaling (Fig. 5e) and functions (Fig. 5f), especially regarding lipid and energy metabolism (such as fatty acid oxidation pathways) were inferred as being inhibited following exposure to HFD. Consistent with this, the analysis predicted as top upstream regulators responsible for the observed transcriptional landscape factors involved in metabolic regulation, such as *Pparg, Hnf4a, Ppara* (Fig. 5g). These were inferred inhibited by the analysis, a prediction confirmed by their observed significant downregulation in the STZ-ablated islets of HFD exposed mice, as compared to the SD mice (Fig. 5g, last column).

Interestingly, in contrast with the DT model, no key cell cycle markers with the exception of some cyclin-dependent kinase inhibitors were observed being regulated in islets from HFD-exposed mice (Fig. 5h). Similarly, only few islet cell markers were deregulated, such as *Slc2a2* (Glut2), *Hnf4a*, or *Arx* (Fig. 5i), while key beta-cell genes like *Ins1, Ins2, Pdx1, Nkx6.1* and *Mafa* did not exhibit a significant regulation.

## The direct comparison of the STZ- and DT-moderate ablation models confirms proliferative differences under HFD

We directly compared the STZ- and DT-moderate ablation (50%) models at ten WPA (weeks post ablation), in SD and HFD conditions, using high-throughput sequencing, followed by pathway analysis (Fig. 6a).

A number of 2005 genes were differentially expressed between the moderate STZ- and DT- ablated islets, under SD conditions (Fig. 6b). Most of these were involved in signaling related to immune system and metabolism, being inferred activated in the STZ-ablated islets, suggesting an increased inflammatory response and metabolism following STZ-induced ablation, as compared to DT-induced ablation (Fig. 6c). This observation was confirmed by the comparison analysis, which indicated opposite activity patterns between the two moderate ablation models for many key metabolic pathways (Fig. 6d). Consistently, the top predicted upstream regulators with observed significant regulation between the conditions were known metabolic regulators (Fig. 6e).

Of note, most key cell cycle regulators and islet cell markers were not significantly deregulated between the two ablation models at 10 WPA, under SD conditions, with few exceptions such as *Cdkn1a* or *Hnf4a* (Fig. 6f). This is in stark contrast with the comparison under HFD, where a large number of key cell cycle signaling was observed in the top, being inferred as inhibited in the STZ-induced partially ablated islets (Fig. 6g). As expected, key cell cycle markers such as *Mki67, Foxm1* and *Aurka* were observed downregulated in the STZ-ablated islets exposed to HFD as compared to their DT counterpart (Fig. 6h). Moreover, the comparison pathway analysis revealed the activation of these pathways specifically in DT-ablated islets exposed to HFD, as compared to their non-ablated HFD-exposed control, in contrast with the equivalent STZ comparison where no activity pattern could be inferred (Fig. 6i). Overall, these data indicate that HFD contributes to the DT-induced proliferation boost, however it does not seem to promote proliferation in the STZ-ablation context.

## Discussions

Here we show that following rapid, moderate 50% DT-induced beta-cell ablation, the pancreatic islet mounts a compensatory response in the absence of an obvious metabolic pressure to regenerate. Early following beta-cell loss, the islet transcriptional landscape incrementally changes, being characterized by a transient activation of inflammation and proliferation signatures at 5 DPA and ultimately exhibiting a recovering beta-cell signature by 30 DPA. By using immunodeficient RIP-DTR mice, we found that, despite their coincidental occurrence, there is no clear causal relationship between the inflammation and proliferation signature. In contrast, additional metabolic stressors, such as dietary challenge, promote beta-cell proliferation in moderately ablated islets, yet have a negative impact on beta-cell signature. Further, we showed that in a second model of moderate (50%) beta-cell ablation, based on STZ administration, the dynamic of the compensatory events is slower, with an observed increase in proliferation being detected as late as 30 DPA. Moreover, in this context, the dietary challenge failed to promote proliferation and induced mostly metabolic changes, suggesting that distinct compensatory mechanisms are

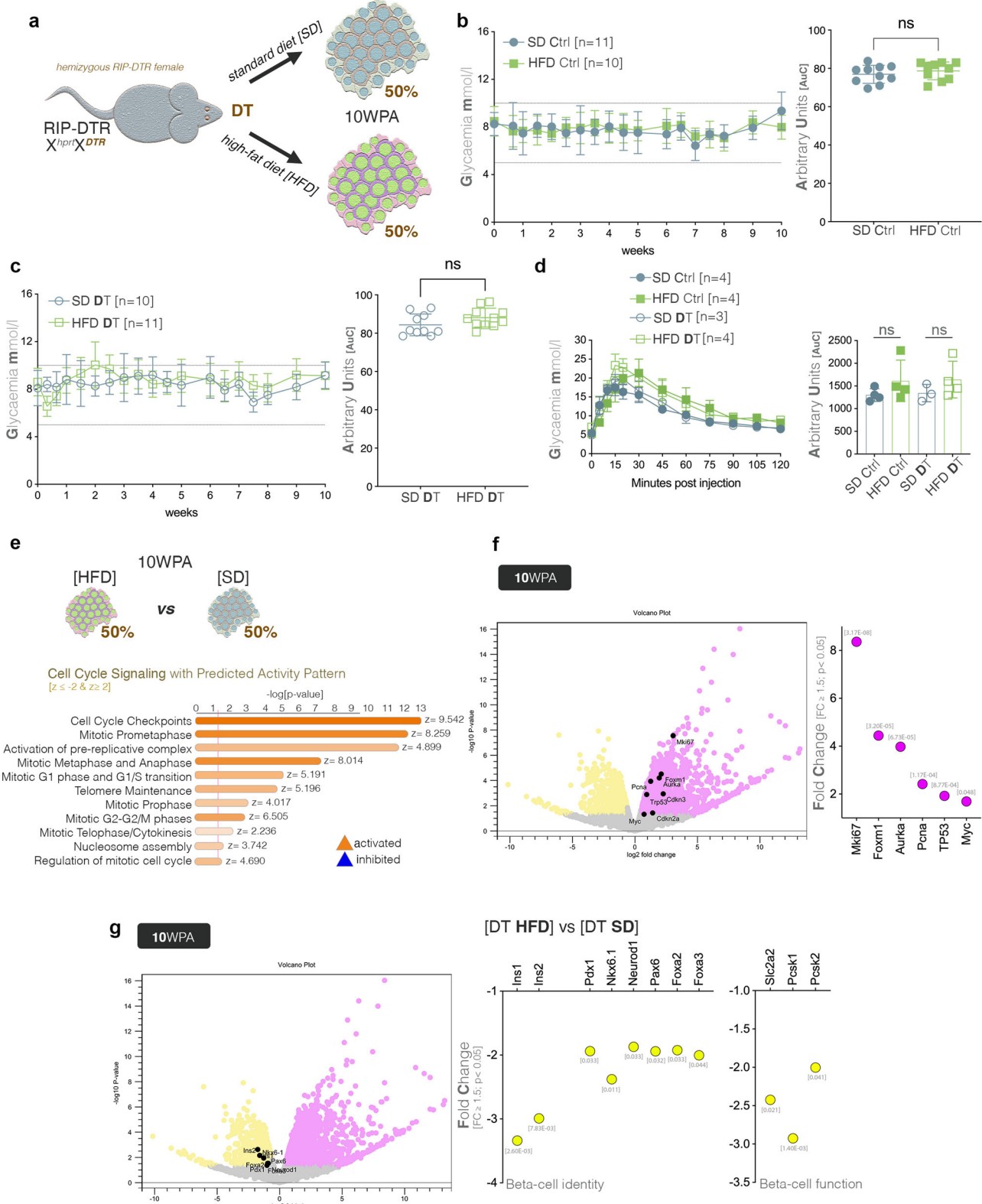

**Fig. 4 | Assessment of the compound stress from diet and partial beta-cell loss on the compensatory mechanisms. a** Schematic illustration of the combinatorial treatment between moderate DT ablation and dietary stress (HFD – high fat diet, SD - control synthetic diet). **b** Glycaemic curves tracking control mice over a 10-week period on either HFD or SD, with accompanying AuC comparisons (Welch's corrected *T* test). **c** Glycaemic curves tracking DT-ablated mice over a 10-week period on either HFD or SD, with accompanying AuC comparisons (Welch's corrected *T* test). **d** Graphs depicting results of IPGTT at 10 WPA in DT-ablated and control mice under RC (*n* = 3 DT, *n* = 4 CTRL) and HFD regimen (*n* = 4 DT, *n* = 4 CTRL)

as well as their respective AuCs. All data (**b**–**d**) are represented as mean ± SEM. **e** Predicted activity pattern of cell cycle pathways in HFD or SD treated cohorts (z-score ≤ -2 [inhibited, blue], z-score ≥ 2 [activated, orange]. **f** Volcano plot and graph displaying the observed upregulation of proliferation markers and cell cycle regulators when comparing mice fed on HFD or SD, *p*-values are listed above bar. **g** Volcano plot and graph displaying the observed downregulation of beta-cell identity and functionality genes when comparing islets from mice fed with HFD or SD, p-values are listed below bar.

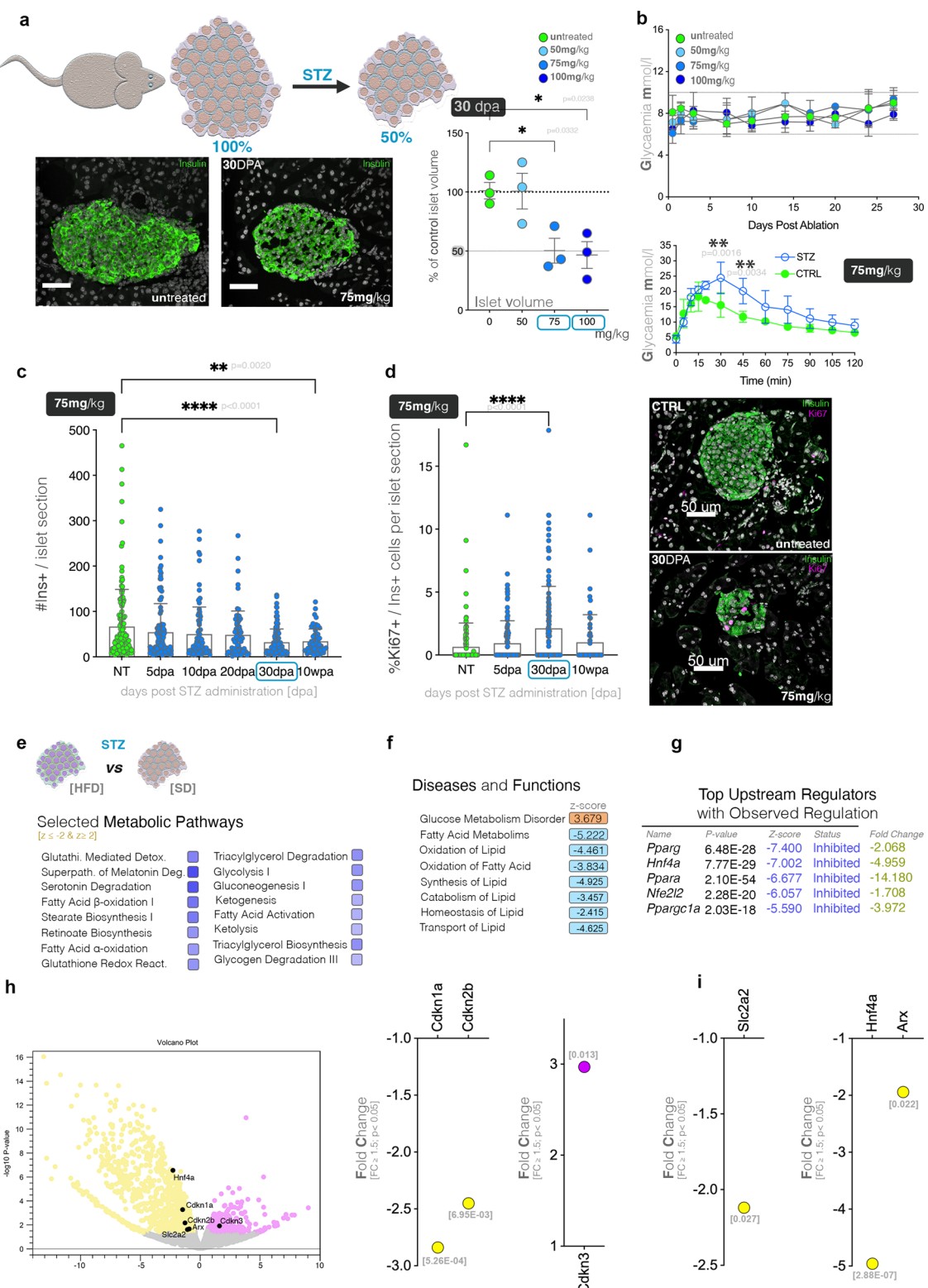

deployed in the two analyzed contexts, despite the similar amount of beta-cell loss.

Previous studies indicated that an amount as little as 10-14% of the original beta-cell mass is enough to maintain normoglycemia in homeostatic conditions[32]. Nevertheless, higher values are required to cope with metabolic stressors, leading to an increased regenerative pressure to mount a compensatory response. This is dependent on the amount of beta-cell loss[10], with total destruction (99.98%) leading to regeneration by transdifferentiation from neighboring endocrine cell types[6–8,12], while sub-total loss (80–90%) being demonstrated to trigger a proliferative response of the surviving beta-cells[13–15,35,36]. However, in both cases, the residual beta-cell mass following loss is below, or around the physiological threshold and thus there will be an increased demand for regeneration and replacement of the lost beta-cells. Therefore, the existence of compensatory mechanisms involving drastic changes of the transcriptional landscape following moderate beta-cell loss, including tissue remodeling and cell proliferation, is

**Fig. 5 | Moderate beta-cell ablation following a single dose of STZ. a** Illustration of the intended efficiency for STZ-induced ablation as well as graph depicting the dose-dependent efficiency of ablation (one-way ANOVA, each data point represents one distinct mouse, an average of 84 islet sections were assessed/mouse) including representative immunofluorescent images (DAPI (white), insulin(green), scale bar – 50 μm). **b** Glycaemic curves showing glycaemia over the 30 days following STZ injection for the various dosages and graph depicting results of IPGTT at 10 WPA in STZ-ablated (n = 3) and control (n = 4) mice (two-way ANOVA with Bonferroni correction). **c** Number of Ins+ cells per islet section up to 10 weeks post STZ-induced ablation following a single 75 mg/kg dose (one-way ANOVA, N = 3 mice, an average of 99 islet sections were counted per condition). **d** Percentage of Ki67 positive insulin cells in control, 5, 30 and 30 DPA following a single dose of 75 mg/kg STZ (one-way ANOVA, N = 3 mice, an average of 60 islet sections were counted per condition) and representative immunofluorescence images of untreated and 30 DPA islets (DAPI (white), insulin (green), Ki67(magenta), scale bar – 50 μm). All data (**a–d**) are represented as mean ± SEM. **e** Predicted activity pattern of selected metabolic pathways between STZ-ablated islets (75 mg/kg exposed to either HFD or SD (z-score ≤ -2 [inhibited, blue], z-score ≥ 2 [activated, orange]). **f** Predicted activity pattern of selected top diseases and functions (z-score ≤ -2 [inhibited, blue], z-score ≥ 2 [activated, orange]). **g** Table of top upstream regulators predicted inhibited and observed downregulated in mice treated with 75 mg/kg and exposed to either HFD or SD. **h** Volcano plot and graph displaying the observed regulation of cell cycle regulators when comparing islets from HFD fed mice to SD fed mice after STZ treatment (75 mg/kg). **i** Observed regulation of islet markers when comparing HFD and SD fed mice after STZ treatment (75 mg/kg).

surprising. This suggests the presence of a highly sensitive molecular "sensor" of beta-cell mass control, initiating compensatory mechanisms even after limited beta-cell loss.

The nature of the compensatory mechanism or mechanisms involved is important from both a fundamental science and clinical perspective, with expansion or functional rebound of the surviving beta-cells being the two most obvious scenarios. The observed increase in beta-cell proliferation in the moderate (50%) DT-ablated islets indicates the presence of a regenerative program characterized by beta-cell expansion. Yet, this is too limited to account for the complete recovery of the beta-cell signature by 30DPA, thus clearly suggesting the occurrence of a functional tunning of the surviving beta-cells. One limitation of this study is the inability to demultiplex how much each mechanism contributed to recovery. This proved to be technically difficult in partially ablated islets, as the variation in islet size caused by the sectioning plane, as well as the inter-islet variations caused by random X-inactivation, makes the quantification of a moderate beta-cell population increase very challenging.

Of note, besides *insulin*, we also observed the significant upregulation of *glucagon (Gcg)* and *somatostatin (Sst)* transcripts at 30DPA, without an increase in alpha- or delta-cell proliferation. Interestingly, *Gcg* upregulation as well as an elevated pancreatic glucagon content and glucagonemia in the absence of alpha-cell proliferation was also reported following total (99.98%) ablation, suggesting it as a common response to beta-cell loss, regardless of ablation extent. It is thus tempting to speculate that following moderate (50%) beta-cell ablation we witness a cell identity reinforcement phenomenon similar to the one we previously described following total (99.8%) ablation. Indeed, following complete beta-cell loss the vast majority of alpha-cells reinforce their alpha-cell signature, which acts as a regenerative brake, preventing their conversion towards a beta-cell fate[6]. It is however important to state that in the current study we did not observe the occurrence of cell conversion phenomena following moderate ablation. Consequently, the reinforcement of alpha-cell signature could serve other purposes in the regenerative processes, which analysis requires further studies at single cell level.

Previous studies demonstrated the impact of inflammation on the regeneration potential in many tissues, organs, and model systems[42,43]. Yet, in moderately (50%) ablated islets, the two processes appeared coextensive, with the 5 DPA proliferation burst still occurring in severely immunodeficient mice suggesting that inflammation is not absolutely required for the regenerative response. Nevertheless, our data indicated an increase in the macrophage signature markers, especially the M2 anti-inflammatory type, even in the immunodeficient NSG background, hinting that these might be implicated in the regenerative response. Yet, with the methods used in this study it is not possible to demultiplex their exact role in the process. An increase in the Cd45+ cells at 5DPA as well as the involvement of macrophages in the regenerative process was also previously observed following total (99.98%) beta-cell ablation[44], suggesting it as another potential common denominator of DT-induced beta-cell loss. Interestingly, macrophages' role in regeneration was already documented in a wide range of regeneration models and organisms[45–50], including other models of beta-cell regeneration[51,52], where they fine-tune the regeneration response, by controlling the transition towards an anti-inflammatory environment[53–56].

Of note, this is in line with the limited inflammation also observed after total beta-cell ablation of the RIP-DTR mice[8], as compared to other beta-cell regeneration models. Based on these observations, it is tempting to speculate that in the RIP-DTR model the regeneration is initiated by the apoptotic beta-cells, similar to the apoptosis-induced compensatory cell proliferation observed in other regenerative systems[57–60]. However, proving this scenario will require targeted line of investigation, which is not the object of this study.

In contrast, we showed here that prolonged exposure to high fat diet (HFD) stimulated the proliferation of moderately (50%) DT-ablated islets, which demonstrated a link between the metabolic demand and regenerative potential in this context. This is in line with previous studies indicating a positive regulation of beta-cell proliferation by HFD[38].

Yet, HFD failed to improve beta-cell signature following moderate ablation, indicating that it has a negative impact on beta-cell functional rebound. Thus, these data suggest a differential action of HFD on the two compensatory mechanisms involved, a positive action on beta-cell proliferation and a negative action on beta-cell functional compensation.

The presence of a late beta-cell proliferation burst (30DPA) in the slow model of STZ-induced beta-cell ablation, suggests it as a common islet response to moderate (50%) ablation. This is particularly important, considering the difference in the ablation temporal dynamic between the DT- and STZ-induced beta-cell ablation. The slow ablation dynamic in the STZ model as compared to DT-induced ablation was previously reported[61,62] and is inherent to its mode of action. Moreover, one should also consider the variable islet ablation efficiency[63], which will increase the variability between the ablated mice, and thus impact the observation of subtle cellular processes, or minute changes in the transcriptional landscape. Last, one more drawback of the STZ-based ablation is the toxicity of the compound, especially for the kidneys, which could systemically and indirectly impact islet functionality.

In contrast with DT-induced moderate (50%) ablation, the HFD administration did not promote proliferation in the STZ-ablation model. This suggests that, although the response is similar (proliferation), the regeneration initiation signal is different, probably due to the very different dynamic of cell death in the two systems (fast versus slow). A future comparison study of the two types of cell death dynamics and the dying cells-related signaling would be able to confirm/reject such a scenario, nevertheless, this would require a different experimental setup than the one employed here.

Along the same lines, one should also consider the limitations of this study, such as the use of bulk transcriptomics, which cannot properly demultiplex the transcriptional response of each islet subpopulation to bet-cell loss. Moreover, by using this approach we cannot completely exclude that relative changes in islet population abundancy might impact the observed changes in gene expression. Thus, further studies using a single cell RNAseq approach will be required for mapping the cellular crosstalk fueling the regenerative response following moderate beta-cell loss.

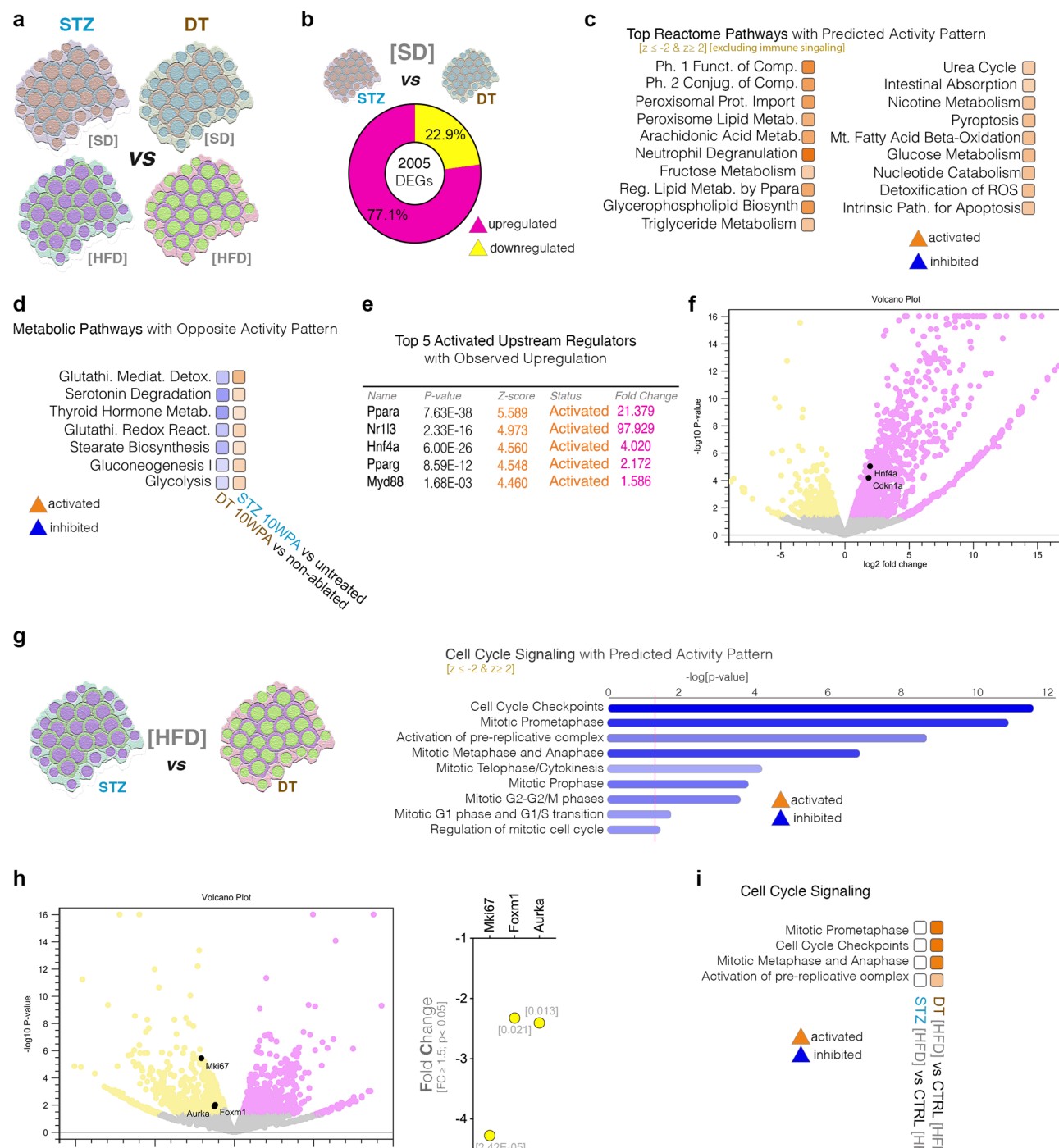

Fig. 6 | Direct comparison between STZ- and DT- moderately ablated islets exposed to either HFD or SD. a Illustration of the comparisons between ablated islet from mice fed either on SD or HFD. b Pie-chart illustrating the number of DEGs (FC ≥ 1.5, p < 0.05) and their regulation between STZ- and DT- ablation on mice exposed to SD. c Top predicted Reactome pathways with predicted activity pattern (z-score ≤ -2 [inhibited, blue], z-score ≥ 2 [activated, orange]) excluding immune pathways) between STZ- and DT-ablated islets from mice fed SD. d Comparison pathway analysis of the DEGs characterizing the DT-ablated and STZ-ablated islets at 10 WPA compared to their respective unablated control (z-score ≤ -2 [inhibited, blue], z-score ≥ 2 [activated, orange]). e Table of predicted activated top upstream regulators with observed upregulation characterizing the direct comparison between

STZ- and DT treated islets in mice fed with SD. f Volcano plot between STZ- and DT treated islets in SD fed mice exhibiting the significant upregulation of Cdkn1a and Hnf1a (g) Predicted activity pattern for cell cycle signaling pathways when comparing STZ- and DT treated islets from mice fed HFD (z-score ≤ -2 [inhibited, blue], z-score ≥ 2 [activated, orange]). h Volcano plot and graph displaying the observed regulation of proliferation markers when comparing islets from STZ- or DT-treated islets of mice exposed to HFD. i Comparison pathway analysis of the cell cycle signaling predicted activity pattern between STZ- or DT-treated islets exposed to HFD normalized on their respective controls (z-score ≤ -2 [inhibited, blue], z-score ≥ 2 [activated, orange]).

## Materials and methods

### Murine models

All animals used in this study were housed in groups of 2 to 5 in internally ventilated cages (IVC II) at 22 °C, with a 12-h light/dark cycle, and free access to food and water. Caretaking was performed according to guidelines set by the Norwegian Animal Research authority and in accordance with the European Union (EU) Directive 2010/63/EU, and experiments were approved in the following FOTS licences: 19800, 25526, 25531, and by the Romanian competent authority (Authorization No. 590/13.01.2021).

The study used female mice of an age younger than 20-weeks at the time of injections. The choice of all female mice was made to replicate findings in the Hemi-DT model, which can only use females due to the reliance on random X inactivation.

For the injection of STZ, we utilized WT mice of a mixed C57BL6/J background. The moderate ablation using DT was achieved with using HTZ female carriers of the RIP-DTR transgene as described in ref. 8. Briefly, the insertion of RIP-DTR into the HPRT locus on the X chromosome allows for expression of DTR in approximately 50% of beta-cells in HTZ individuals, as determined by the RIP promoter, and random X inactivation. The transgene was introduced on a mixed C57BL6/J background. The NSG RIP-DTR was obtained as previously described[64].

### STZ and DT administration

The STZ injection vehicle was made by dissolving Sodium Citrate buffer in 9% NaCl solution to achieve a final concentration of 0.1 M Citrate buffer with a pH of 4.5. STZ (Sigma) was dissolved in this solution shortly before injection to avoid decay. Dosage was calculated by bodyweight, and only a single dose was administered per mouse. Prior to STZ-injection, mice were fasted for four hours.

The administration of DT was performed through repeated injections as previously described[8,65].

### Glycaemia, Glucose tolerance testing and bodyweight

Glycaemia and bodyweight were measured bi-weekly following IP injection by using a Contour XT glucometer (Bayer) and a tabletop scale. Glucose tolerance was evaluated using IPGTT at dose of 2.0 g/kg of D-glucose (Sigma), following 15 h of fasting. The dynamics of glycaemic levels were followed by repetitive glycaemic measurements spread over two hours after injection.

### High fat diet

Mice were fed either regular chow (RC; cat. No. 824050; 10% AFE fat, SDS) or HFD (cat. No. 824054; 60% AFE fat, SDS) for the duration of the 10-week period. Mice were bi-weekly monitored for changes in bodyweight and glycaemia.

### Immunofluorescence and image analysis

Collection and preparation of mouse pancreases was previously described in ref. 66. Tissue blocks were cut into 10 μM sections (Leica CM1950 Cryo-microtome, Leica) and mounted on SuperFrost Plus slides (Thermo Scientific). The antibodies used were as follows: guinea pig anti-insulin (1:400, AE804[67] Geneva Antibody Facility), mouse IgG1 anti-glucagon (1:1000, G2654[68] Sigma-Aldrich), mouse IgG anti-Ki67 (1:1000, ab279653[69] Abcam), rabbit anti-somatostatin 28[70] (1:400, ab111912 Abcam), chicken anti-insulin (LS-c96116, LSBio), guinea pig anti-glucagon (1:200, AK247[71] Geneva Antibody Facility), chicken anti-somatostatin 28 (1:400, 366006[72] Synaptic Systems), rabbit anti-insulin (1:200, 15848-1-AP[73] Thermo Scientific), and rat anti-CD45 (1:100, NB100-77417, Novus Biologicals). For epitope retrieval of nuclear targets (Ki67), heat induced epitope retrieval was performed in 10 mM citrate buffer pH 6.0 for 5 min using a pressure cooker (Ninja).

Visualization of the primary stains was performed with the following secondary antibodies from Invitrogen: Goat anti-guinea pig A488(A11073), donkey anti-rabbit A546(A10040), Goat anti-chicken A594(A11042), goat anti-chicken A488(A11039), goat anti-mouse A647(A21235), goat anti-mouse IgG1 A647(A21240), goat anti-rat A647 IgG (A21247). All the secondary antibodies, along with nuclear staining with DAPI were used at a concentration of 1:500. The stained slides were mounted using Aqueous mounting media (Abcam).

Two image acquisition systems were used in this study, the Leica TCS SP8 STED 3x (Leica Microsystems) and Andor Dragonfly 5050 confocal microscope (Oxford Instruments). The LASX software (Leica) was used for the manual counting of cells. Segmentation of insulin volumes were performed at a 2,0 μm resolution in the Imaris software (Bitplane). For the automated supervised counting of DAPI nuclei, a macro using Auto Threshold feature with the Otsu Dark method in FIJI was used[74]. A region of interest (ROI) was drawn manually to define the islet area from the generated mask before cells were counted by "Automatic Particle counting" while excluding particle with an Area of less than 10 μm².

### Islet isolation and RNA extraction

To generate islet samples for transcriptomic analysis, pancreatic islets from 3–6 mice per condition were isolated through injection of a 0.2% collagenase (Sigma) solution into the ampulla of Vater, with the common bile duct clamped. Islets were separated from the exocrine by nine minutes of digestion in a water bath at 37 °C, followed by separation in a Histopaque (Sigma Aldrich) gradient consisting of phases at 1.080, 1.100, 1.110, and 1.119 g/ml. The islets were manually recovered from the gradient and hand-picked using a light-microscope to avoid exocrine contamination. At the end of islet isolation, the islet were stored in RLT-buffer (Qiagen) with added β-Mercaptoethanol (Sigma Aldrich), to stabilize RNA before extraction. RNA was extracted from the islet samples using the RNeasy Mini Kits (Qiagen). The purified RNA was assessed using the Tapestation 4150 (Agilent) to establish RNA amount and quality, or by NanoDrop 2000c (ThermoFisher Scientific), before shipment to the Qiagen Genomics or Novogene facilities for sequencing.

### Sequencing, data and pathway analysis

All samples underwent secondary quality control, and library preparation at the at the Qiagen genomics faculty prior to sequencing. The exception is the sequencing data from islets of the NSG RIP-DTR mice, which were performed by Novogene. The output sequencing files were processed using the CLC Genomics Workbench (24.0.0), using the "Trim Reads Tool", where sequences are trimmed based on quality stores and nucleotide ambiguity (Max ambiguity = 2). Trimmed reads were mapped using the "RNAseq Analysis" tool, aligning to the GRCm39 genome. Differential expression comparisons were performed with either the "Differential Expression for RNAseq" tool or "Differential Expression in Two groups", before uploading the datasets (Supplementary Data 1) to Ingenuity Pathway Analysis[75] (IPA, Qiagen). Pathway analysis were performed on DEG lists after filtering (FC ≥ 1.5, $p < 0.05$), with network size set to 35, including all data sources from mouse species, and including all entities except chemical subclasses.

### Statistical analyses

Statistical analyses were performed using GraphPad Prism v9.5.1 (Graph-Pad Software Inc., USA). To assess the statistical differences between groups for the immunofluorescence data quantification and physiological parameters we used one-way ANOVA test, unless otherwise specified in the corresponding figure legend. Area under the curve was calculated for the glucose tolerance and long-term glycemia curves using the trapezoidal rule. In figures, data are represented as mean ± SEM (standard error of the mean) unless otherwise specified (Supplementary Data 2). Statistical significance was defined at $P < 0.05$ (∗), $P < 0.01$ (∗∗), $P < 0.001$ (∗∗∗), and $P < 0.0001$ (∗∗∗∗).

### Reporting summary

Further information on research design is available in the Nature Portfolio Reporting Summary linked to this article.

## Data availability

The datasets were uploaded to the NCBI's Gene Expression Omnibus[76] GEO database and are available under to following identifiers GSE26517 and GSE26518. All other data are available from the corresponding author on reasonable request.

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

## Acknowledgements
We thank J.H. Gudmestad for technical help, A. Altankhuyag for animal care and H.A. Dale for Imaris assistance. Confocal imaging was performed at the Molecular Imaging Center (MIC), Department of Biomedicine, University of Bergen. This work was supported by funds from the Research Council of Norway (NFR 304615 and 314397), Novo Nordic Foundation (NNF21OC0067325), Diabetesforbundets forskningsfond and University of Bergen to S.C.; NO Grants 2014-2021, under Project contract no. 21/2020 (RO-NO-2019-0544; BETAUPREG) to A.V. and S.C.; A.F.M. and L.U. were supported by doctoral fellowships from the Faculty of Medicine, University of Bergen. The funding sources had no role in the study design, its execution, analyses, interpretation of the data, nor the decision to publish these results.

## Author contributions
A.F.M., L.U., and S.C. analysed the transcriptomics data; A.F.M., U.L., and N.K. performed the mouse work and collected the physiological data, L.M.D. and A.M.V. prepared samples from NSG RIPDTR mice; A.F.M., U.L., N.K., and L.U. performed fluorescence imaging and analysis; L.G., S.C., A.F.M. conceived the experiments and interpreted the observations, A.-M.V., L.G., and S.C. supervised the work, A.F.M., and S.C. wrote the manuscript, P.L.H., A-M.V., and L.G. edited the manuscript. All authors approved the final version of the manuscript.

## Funding

## Competing interests

The authors declare no competing interests. The authors declare that the research was conducted in the absence of any commercial or financial relationships that could be construed as a potential conflict of interest. Simona Chera is an Editorial Board Member for Communications Biology, but was not involved in the editorial review of, nor the decision to publish this article.
