## [Peer Review File · Communications Biology]

Reviewers' comments:

Reviewer #1 (Remarks to the Author):

This paper studies the mechanisms of beta cell regeneration following ablation, and the effect of metabolic stressors, such as diet, in this response. The paper provides a comprehensive examination on beta-cell regeneration using a 'moderate' beta cell loss model where ~50% of the beta cell mass is destroyed. The paper combines bulk transcriptomics, immunohistochemistry, and imaging of mouse islets to tackle the regeneration process. The paper also uses multiple mouse models to study specific effects (such as inflammation) that could be potentially driving regeneration itself, and compares it to other environmental factors such as diet. The paper is well written, easy to follow, and it presents results that are of interest to the community, however, there are some aspects that need to be improved and clarified.

Main questions

- 1) In Figure 1a, it is unclear how many islets / mice were analyzed in each condition? Are these averages of multiple islets for 3 animals? As it is explained now, it could look like if only 3 islets were studied, which is a very small number and would require improvement to mitigate potential bias. Same for Figure 1b, how many animals were included in the generation of the glycaemic curves?
- 2) In Figure 1c, the CTRL group has consistently lower values between 15- and 60-minutes post-injection (especially at t=20min). Do the authors have an explanation for this?
- 3) In Figure 3, changes in gene expression in bulk islets at different time points are presented. The authors identify an increase in inflammation at days 3 to 5, followed by an increase in beta-cell markers at day 30. Given that most of the work is performed using bulk transcriptomics, it is hard to disentangle effects that could be driven by relative changes in population abundances (loss of beta cells, immune cells), and those that reflect true changes in gene expression in beta cells (or other islet cells). For instance, the extent of induced apoptosis (~ 50% of beta cells), could alone lead to inflammation, and recruitment of neutrophils to clear apoptotic cells. Similarly, after depletion of 50% of beta cells, the relative abundance of alpha cells, and delta cells will be increased, leading to apparent log fold changes in pathways and genes relative to these cell types. It would be useful to discuss in more detail how these potential confounding effects are addressed, or at least how they could influence the main findings.
- 4) Related to the previous point, in Figure 2f and in the main text, the authors briefly mention the upregulation of SST and GCG in the cells at 30 DPA compared to 5 DPA, yet they do not offer any explanation for this observation. Can the authors comment on this?
- 5) The visualizations in Figure 2 and 3 with individual log-fold changes between pairwise conditions is sometimes hard to follow. Line plots across the longitudinal axis could help visualize relative changes across all time points (e.g. using as a reference D0).
- 6) The authors observe the upregulation of "key beta-cell" markers in 30 DPA and they speculate about a potential "regenerative event". However, there is no obvious increase in beta cell mass in Figure 1a? Can the authors comment on this? Would sampling of more islets show this increase?
- 7) In Figure 3g, I recommend the authors to sample a substantial number of islets per condition. Can the authors provide the total number of islets/sections studied? Also, while they provide an image of an islet at 30 DPA, this condition is not included in the quantitative boxplot.
- 8) In line 156, could the authors clarify the sentence "Overall, these data indicate a transient

proliferation wave peaking at 5 DPA, consistent with a regeneration-based compensatory event.". Specifically, what makes it a wave?

9) The use of an immunodeficient mouse model is sound, but NSG mice have intact neutrophils and macrophages that play a crucial role in the clearing the apoptotic beta cells which drive the inflammation in these models. Could the authors comment on this? The authors could also consider adding an inflammation component at 5-DPA by LPS injection or Poly I:C, to further validate their hypothesis.

10) Previous studies have established that a high-fat diet (HFD) induces proliferation of beta cells. Please include a citation for reference when discussing this topic in lines ~174-178.
Chen C, Chmelova H, Cohrs CM, Chouinard JA, Jahn SR, Stertmann J, Uphues I, Speier S. Alterations in β -Cell Calcium Dynamics and Efficacy Outweigh Islet Mass Adaptation in Compensation of Insulin Resistance and Prediabetes Onset. *Diabetes*. 2016 Sep;65(9):2676-85. doi: 10.2337/db15-1718. Epub 2016 Apr 8. PMID: 27207518.

11) In Figure 4b authors only show glycaemic curves and no IPGTT. It might be useful to also show IPGTT in these conditions.

12) In Figure 5a, it appears that only three islets were analysed per condition. I have the same concern as with Figure 1a.

13) Could the authors provide an explanation for why all examples of islets at 30 DPA appear to be considerably smaller than those at other time points, even if ablation mostly happens much earlier on?

Other comments

Figure 1 precedes the explanation of DPA in the text. It would be helpful to introduce this term earlier to provide context for the reader.

The concept that 50% ablation of beta cells is 'moderate' might be controversial, and would benefit of being placed in context of previous studies with extreme ablation of beta cells.

Reviewer #3 (Remarks to the Author):

This study aimed to investigate the interplay between pancreatic beta cell loss and islet regeneration/proliferation by using genomic sequencing. Two different animal models were used including DT-induced rapid beta cell ablation and STZ-induced slower beta cell ablation. In addition, the authors tried to find out the roles of inflammation and dietary stress in this process. Immunodeficient mice and HFD were used to understand how inflammation and metabolic stress affected the beta cell proliferation. The results showed that in the scenario of acute beta cell ablation, beta cell proliferation response occurred independently of inflammation, and exposure to HFD stimulated beta cell proliferation but negatively impacted cell function. While in slow beta cell ablation, a delayed but similar proliferative response was identified. But HFD did not promote proliferation. This is an interesting finding. The reviewer has the following comments and questions:

1) The study demonstrated the proliferation of beta cells in both models. Did the authors see an improvement of islet function over time along with beta cell regeneration in both models?

2) The authors should demonstrate what is the significance of beta cell proliferation with histopathological staining.

3) The authors only focused on beta cell, how about alpha cells, delta cells, etc? Along with beta cell proliferation, what is the change of alpha cells?

4) Transdifferentiation is another important process. Did authors see any signals of pancreatic cell transdifferentiation upon beta cell ablation?

Point-by-point Reply

We are very grateful to both Reviewers for their positive thoughts and proposed suggestions on our manuscript.

Reviewer #1 (Remarks to the Author):

This paper studies the mechanisms of beta cell regeneration following ablation, and the effect of metabolic stressors, such as diet, in this response. The paper provides a comprehensive examination on beta-cell regeneration using a 'moderate' beta cell loss model where ~50% of the beta cell mass is destroyed. The paper combines bulk transcriptomics, immunohistochemistry, and imaging of mouse islets to tackle the regeneration process. The paper also uses multiple mouse models to study specific effects (such as inflammation) that could be potentially driving regeneration itself, and compares it to other environmental factors such as diet. The paper is well written, easy to follow, and it presents results that are of interest to the community, however, there are some aspects that need to be improved and clarified.

1) In Figure 1a, it is unclear how many islets / mice were analyzed in each condition? Are these averages of multiple islets for 3 animals? As it is explained now, it could look like if only 3 islets were studied, which is a very small number and would require improvement to mitigate potential bias. Same for Figure 1b, how many animals were included in the generation of the glycaemic curves?

We thank the reviewer for pointing out this omission. Indeed, for Figure 1a these are averages of multiple islets for 3 animals (each dot represents one animal, an average of 55 islet sections was quantified / animal). For Figure 1b each data point represents the mean glycemia of several mice, DT (n=10) and CTRL (n=11). **We now added this info in the legend** (for Figure 1a-c) as well as in the Figure 1b and Figure 1c panels.

2) In Figure 1c, the CTRL group has consistently lower values between 15- and 60-minutes post-injection (especially at t=20min). Do the authors have an explanation for this?

We indeed noticed this apparent tendency. However, neither the total area under curve (AuC) nor the AuC comparison of the 15 to 60 min interval did not reveal any significant difference (see Reviewer Figure 1a). This was also true for the two-way Anova timepoint comparisons (see Reviewer Figure 1b). Moreover, direct glycemia comparison at 20 min did not reveal a significant difference with either parametric or non-parametric tests (see Reviewer Figure 1c). Yet, we agree with the reviewer that this question might arise from the readers and thus **we have now also included several of these graphs in the Figure 1c and amended the legend accordingly.**

Reviewer Figure 1. a) Area under curve (AuC) for the total IPGTT course and 15 to 60 subperiod; **b)** two-way Anova timepoints comparison. **c)** glycemia comparison at 20min timepoint (non-parametric and parametric test)

3) In Figure 3, changes in gene expression in bulk islets at different time points are presented. The authors identify an increase in inflammation at days 3 to 5, followed by an increase in beta-cell markers at day 30. Given that most of the work is performed using bulk transcriptomics, it is hard to disentangle effects that could be driven by relative changes in population abundances (loss of beta cells, immune cells), and those that reflect true changes in gene expression in beta cells (or other islet cells). For instance, the extent of induced apoptosis (~ 50% of beta cells), could alone lead to inflammation, and recruitment of neutrophils to clear apoptotic cells. Similarly, after depletion of 50% of beta cells, the relative abundance of alpha cells, and delta cells will be increased, leading to apparent log fold changes in pathways and genes relative to these cell types. It would be useful to discuss in more detail how these potential confounding effects are addressed, or at least how they could influence the main findings.

We thank the reviewer for pointing this out. Indeed, the use of bulk transcriptomics is a clear limitation of the study and, as requested, **we have now included a paragraph about this limitation in the Discussion.**

However, we would like to add, that if the change in the population abundances will be an important issue with our RNAseq pipelines, one would expect an extreme alpha-cell markers inflation especially following total beta-cell ablation (99.98% - for the model see *Thorel et al., Nature, 2010; Chera et al., Nature, 2014*), which does not seem to be the case (please see below Reviewer Figure 2 from our published datasets). The fold change of *Gcg*, *Arx*, *Mafb* transcripts in ablated (5DPA) versus non-ablated control are similar in isolated islets and isolated alpha cells. This should not be the case if an important artificial inflation would occur.

Reviewer Figure 2. Fold change of *Gcg*, *Arx* and *Mafb* in a) isolated islets and b) isolated alpha cells (generated from dataset in Oropeza et al., 2021)

In addition to the explanation above, we would respectfully like to add that we almost exclusively compared ablated points along the manuscript (including in Figure 3) with very few

exceptions thus the relative abundance of the other endocrine non-beta cell types should be rather comparable between conditions.

Nevertheless, **we entirely agree that one cannot completely exclude an interference from the immune and other cell populations' signature, and we now discuss this possibility in the Discussion.**

4) Related to the previous point, in Figure 2f and in the main text, the authors briefly mention the upregulation of SST and GCG in the cells at 30 DPA compared to 5 DPA, yet they do not offer any explanation for this observation. Can the authors comment on this?

We thank the reviewer for pointing out this omission. Interestingly, the increase in glucagon transcription was also reported following total ablation along with significantly elevated glucagonemia and pancreatic glucagon content (see Supplemental Figure S5 in *Thorel et al., Nature, 2010*) suggesting that it is a common response to beta-cell loss. As in both cases (50% moderate ablation and 99.98% total ablation) there was no significant increase in alpha-cell proliferation (see this manuscript and, respectively, *Thorel et al., Nature, 2010*), the most probable scenario is an increase of glucagon transcription rate in the alpha-cells.

It is tempting to speculate that we observe here the same cell identity reinforcement phenomenon that we previously reported for alpha-cells following total (99.98%) beta-cell ablation (*Cigliola, Ghila, Thorel et al., Nature Cell Biology, 2018*). In that paper we showed that following complete beta-cell loss the vast majority of alpha-cells reinforce their alpha-cell signature, which acts as a regenerative brake, preventing their conversion towards a beta-cell fate. **We now inserted in the manuscript graphs indicating the ratio of proliferating (Ki67+) alpha- and delta-cells** (in revised Supp.Fig 1k) and **added in the Discussion the possibility of a cell identity reinforcement phenomena similar to what was previously described for total beta-cell ablation.**

5) The visualizations in Figure 2 and 3 with individual log-fold changes between pairwise conditions is sometimes hard to follow. Line plots across the longitudinal axis could be help visualize relative changes across all time points (e.g. using as a reference D0).

As we focused on specific intervals along the regenerative timeline, such as (5 DPA and 3 DPA) or (30 DPA vs 5 DPA) we believe that displaying all time points will make the graphs even more confusing.

Regarding the relative changes display there are two issues. First, the relative abundance varies tremendously between the different markers, which makes it impossible to display in the same graph, while other markers completely superimpose. Second, as we used the Qiagen CLC Genomics pipeline for analysis, the fold changes are calculated from the generalized linear model, which corrects for differences in library size between the samples and the effects of confounding factors. Thus, the simple display of the normalized counts will not allow a correct visualization of the relative changes (it is not possible to derive these fold changes from the original counts by simple algebraic calculations). **Nevertheless, we appreciate reviewer's point, and we now added volcano plots to increase readability, while also keeping the old format for wet-lab readers expecting qPCR-like visualization.**

6) The authors observe the upregulation of "key beta-cell" markers in 30 DPA and they speculate about a potential "regenerative event". However, there is no obvious increase in beta cell mass in Figure 1a? Can the authors comment on this? Would sampling of more islets show this increase?

We thank the reviewer for pointing this out, and we apologize for the rather lengthy explanation below:

The lack of observable significant increase in beta cells numbers after only 50% ablation is unfortunately expected. The main problem is the survival of a significant fraction of the beta-cell population, which masks the very limited regenerative/proliferative events. While regeneration is also limited following total beta-cell ablation, the lack of surviving beta-cells makes it easily observable. In contrast, in 50% ablation the inherent large variations of the surviving beta-cell population make the identification of small population increases very difficult. The combination of different sectioning planes (this is not a problem when no beta-cells are left) and random X-inactivation inter-islet variations (the decrease in the total beta-cell population is indeed ~50%, yet in individual islets the ablation ranges between 40-60%) greatly contribute to this variation. To reach a statistical significance for such a limited regeneration event, considering at least 20% (if all the islets will be sectioned in the same plane) beta-cell population per islet variations, we expect that thousands of islets need to be counted for counteracting this kind of innate variability. Thus, **although we entirely agree with reviewer that counting more islets will eventually reveal a difference**, we believe this will require sacrificing a very high number of animals for a limited gain. **These considerations are included in the discussion as a limitation of the setup/study.**

Nevertheless, a workaround is comparing large islets of similar sizes (an average of 140 cells/islet section), as it partially counteracts the sectioning plane variability by removing the high variability caused by small islets (which frequently represent sectioned islet “poles”, thus with very variable ratios between islet cells populations). This comparison indicates a slight, however significant increase in insulin+ cells between 5 DPA and 10 WPA (see Reviewer Figure 3). **We have now included this quantification in the manuscript**, in the revised Figure 3f.

Reviewer Figure 3. Ratio of insulin+ cells from total islet cells (DAPI)

7) In Figure 3g, I recommend the authors to sample a substantial number of islets per condition. Can the authors provide the total number of islets/sections studied? Also, while they provide an image of an islet at 30 DPA, this condition is not included in the quantitative boxplot.

We apologise to the reviewer for these omissions. **We have now improved the legend clarity** (an average of 68 islet sections were counted per condition) **and included the 30 DPA timepoint** (revised Figure 3e).

8) In line 156, could the authors clarify the sentence “Overall, these data indicate a transient proliferation wave peaking at 5 DPA, consistent with a regeneration-based compensatory event.”. Specifically, what makes it a wave?

The reason for calling it a wave is its transient nature and apparent synchrony at 5 DPA. The proliferation markers and signalling increase towards 5 DPA (Figure 1a-f – compared to 3

DPA and Supp.Figure 1j – compared to 0 DPA) and then decreases between 5 DPA and D30 (Figure 3d, Figure 3e). However, we agree that we didn't prove the synchronicity of the event, but just its transient nature, thus ***we have now reformulated the phrase in the revised manuscript*** and used the term “proliferation event”.

9) The use of an immunodeficient mouse model is sound, but NSG mice have intact neutrophils and macrophages that play a crucial role in the clearing the apoptotic beta cells which drive the inflammation in these models. Could the authors comment on this? The authors could also consider adding an inflammation component at 5-DPA by LPS injection or Poly I:C, to further validate their hypothesis.

We performed IF and quantifications for immune cell markers, such as CD45 (all nucleated hematopoietic cells). We observed an increase of CD45+ cells within the islets at 5DPA, similarly to what we described in *Oropeza et al., BMC Genomics, 2021* in the total ablation model, at the same timepoint. ***We have now included in the manuscript these data*** in Supplemental Figure 1e (IF & quantifications).

Moreover, we have now mined the NSG (immunodeficient) RIP-DTR datasets for neutrophil and macrophage markers. We could not detect any deregulation of neutrophil specific markers, nevertheless we detected a macrophage signature compatible with M2 (anti-inflammatory) and not M1 (pro-inflammatory) macrophages. ***We have now included these results in the manuscript*** (text, Supp.Figure2 and revised Figure 3i, j). In contrast with the immunocompetent model, our Romanian collaborators did not detect an increase in the Cd45+ cells following ablation in the NSG mice,

Regarding the proposed LPS injection or Poly I:C injection, unfortunately, we do not have the authorisation for performing these experiments in mice and obtaining one is very lengthy in Norway (several months).

10) Previous studies have established that a high-fat diet (HFD) induces proliferation of beta cells. Please include a citation for reference when discussing this topic in lines ~174-178. Chen C, Chmelova H, Cohrs CM, Chouinard JA, Jahn SR, Stertmann J, Uphues I, Speier S. Alterations in β -Cell Calcium Dynamics and Efficacy Outweigh Islet Mass Adaptation in Compensation of Insulin Resistance and Prediabetes Onset. *Diabetes*. 2016 Sep;65(9):2676-85. doi: 10.2337/db15-1718. Epub 2016 Apr 8. PMID: 27207518.

We thank the reviewer for this suggestion, ***we have now included this citation in the indicated paragraph and in the Discussion***.

11) In Figure 4b authors only show glycaemic curves and no IPGTT. It might be useful to also show IPGTT in these conditions.

We thank the reviewer for this suggestion, *we have now included the IPGTT of the four conditions analysed* in Figure 4 (revised Figure 4d).

12) In Figure 5a, it appears that only three islets were analysed per condition. I have the same concern as with Figure 1a.

We thank again the reviewer for pointing out this legend omission. As in Figure 1a, in Figure 5a each dot actually represents one mouse and an average of 84 islet sections was quantified / animal. ***We have now added this info in the legend***.

13) Could the authors provide an explanation for why all examples of islets at 30 DPA appear to be considerably smaller than those at other time points, even if ablation mostly happens much earlier on?

The size of the islet at 30DPA is indeed considerably smaller than in non-treated control and 3DPA, however similar in size with 5DPA (in both conditions the islets contain around 50% insulin+ cells - see graphs in Figure 1a). However, in Figure 3e, in the 5DPA condition the apparently larger islet represents actually two islets in very close proximity (giving the wrong appearance of a full-size islet – see below Reviewer Figure 4 for a Sst/Gcg staining clarifying this issue). We selected this picture as both islets are extremely representative, both displaying the exact mean increase in Ki67+ cells displayed in the graph; however, we agree it can create confusion, thus now we replaced it with a clearer image.

Reviewer Figure 4. 5DPA islets (from Figure 3e) in close proximity (red - glucagon, blue - somatostatin, green - insulin, magenta - Ki67).

Other comments

Figure 1 precedes the explanation of DPA in the text. It would be helpful to introduce this term earlier to provide context for the reader.

We thank the reviewer for pointing out this. ***We have now implemented the term explanation earlier in the text and also in the figure and figure legend.***

The concept that 50% ablation of beta cells is ‘moderate’ might be controversial and would benefit of being placed in context of previous studies with extreme ablation of beta cells.

We thank the reviewer for this suggestion. ***We have now addressed in the introduction the notion of moderate ablation in the context of extreme (100%) as well as sub-extreme (70-90%) beta-cell ablation.***

Reviewer #3 (Remarks to the Author):

This study aimed to investigate the interplay between pancreatic beta cell loss and islet regeneration/proliferation by using genomic sequencing. Two different animal models were used including DT-induced rapid beta cell ablation and STZ-induced slower beta cell ablation. In addition, the authors tried to find out the roles of inflammation and dietary stress in this process. Immunodeficient mice and HFD were used to understand how inflammation

and metabolic stress affected the beta cell proliferation. The results showed that in the scenario of acute beta cell ablation, beta cell proliferation response occurred independently of inflammation, and exposure to HFD stimulated beta cell proliferation but negatively impacted cell function. While in slow beta cell ablation, a delayed but similar proliferative response was identified. But HFD did not promote proliferation. This is an interesting finding. The reviewer has the following comments and questions:

1) The study demonstrated the proliferation of beta cells in both models. Did the authors see an improvement of islet function over time along with beta cell regeneration in both models?

The latest timepoint we checked in this study was at 10 weeks post ablation. At this moment the glycemia and IPGTT were similar between ablated mice and their age-matched nonablated controls in DT-treated animals, regardless of diet type. In contrast, despite being normoglycemic, the STZ-treated animals displayed impaired glucose stimulated insulin secretion at the same timepoint (10wpa). However, as the entire timeline is delayed in the STZ mice (proliferation burst at 30 DPA instead of 5 DPA) is it possible that these mice require longer recovery time. **We now included the IPGTTs in the revised manuscript (DT in revised Figure 4d and STZ in revised Figure 5b).**

2) The authors should demonstrate what is the significance of beta cell proliferation with histopathological staining.

The inherent large variations of the surviving beta-cell population, caused by the combination of different sectioning planes and random X-inactivation inter-islet variations (the decrease in the total beta-cell population is ~50%, yet in individual islets the ablation ranges between 40-60%), makes the quantification of the very limited regeneration increase challenging by standard meanings. In contrast, while regeneration is also limited following total beta-cell ablation, the lack of surviving beta-cells facilitates their observation.

Nevertheless, assessing the insulin+ cells in large islet sections (an average of 140 cells/islet section) over time, partially counteracts these problems, by removing the high variability caused by small islets. This analysis indicated a slight, however significant, increase in insulin+ cells between 5DPA and 10WPA. **We have now included this quantification in the manuscript,** in the revised Figure 3f.

3) The authors only focused on beta cell, how about alpha cells, delta cells, etc? Along with beta cell proliferation, what is the change of alpha cells?

We thank the reviewer for pointing out this. **We have now included in the revised version of the manuscript data regarding alpha- and delta-cells.** Briefly, we observed the upregulation of both *Sst* and *Gcg* transcripts between 5 and 30 DPA (**revised Supp.Figure 1g**), without an increase in the proliferation rate of either alpha- or delta-cells (**Supp.Figure 1k**). The upregulation in the *Gcg* transcripts was also observed following total (99.98%) beta-cell ablation (Supp.Figure S5 in *Thorel et al., Nature, 2010*) coupled with significantly elevated glucagonemia and pancreatic glucagon content in the absence of increased proliferation. **We have now also discussed this in the manuscript.**

4) Transdifferentiation is another important process. Did authors see any signals of pancreatic cell transdifferentiation upon beta cell ablation?

We did not observe an increase alpha-to-beta conversion in this context (by quantifying bihormonal cells using insulin and glucagon staining). We have now added these data as a graph in revised Supp.Figure 1i.

REVIEWERS' COMMENTS:

Reviewer #1 (Remarks to the Author):

The authors have satisfactorily addressed most of my comments and provided a reasonable explanation for the experiments that could not be completed within the given timeframe. I find the paper to be novel and believe it contains results of significant interest to the islet research community. Therefore, I recommend its publication.

Reviewer #3 (Remarks to the Author):

The authors has honestly and fairly addressed all comments and questions raised by the reviewer #3. The answers made sense to the reviewer. So the reviewer has no additional questions.